# Metabolite Fingerprinting for Identification of *Panax ginseng* Metabolites Using Internal Extractive Electrospray Ionization Mass Spectrometry

**DOI:** 10.3390/foods12061152

**Published:** 2023-03-09

**Authors:** Xueyan Yuan, Xiaoping Zhang, Jiaquan Xu, Jianhua Ye, Zhendong Yu, Xinglei Zhang

**Affiliations:** 1Jiangxi Key Laboratory for Mass Spectrometry and Instrumentation, East China University of Technology, Nanchang 330013, China; 2School of Nuclear Science and Engineering, East China University of Technology, Nanchang 330013, China

**Keywords:** ginseng, metabolites, internal extractive electrospray ionization mass spectrometry, sequential sampling, metabolite fingerprinting

## Abstract

Ginseng, a kind of functional food and medicine with high nutritional value, contains various pharmacological metabolites that influence human metabolic functions. Therefore, it is very important to analyze the composition and metabolites of ginseng. However, the analysis of active metabolites in ginseng samples usually involves various experimental steps, such as extraction, chromatographic separation, and characterization, which may be time-consuming and laborious. In this study, an internal extractive electrospray ionization mass spectrometry (iEESI-MS) method was developed to analyze active metabolites in ginseng samples with sequential sampling and no pretreatment. A total of 44 metabolites, with 32 ginsenosides, 6 sugars, and 6 organic acids, were identified in the ginseng samples. The orthogonal partial least-squares discriminant analysis (OPLS-DA) score plot showed a clear separation of ginseng samples from different origins, indicating that metabolic changes occurred under different growing conditions. This study demonstrated that different cultivation conditions of ginseng can be successfully discriminated when using iEESI-MS-based metabolite fingerprints, which provide an alternative solution for the quality identification of plant drugs.

## 1. Introduction

Ginseng is the root of the perennial herb *Panax ginseng* (Araliaceae), which is mainly distributed in the three provinces of northeast China [1]. Ginseng is a strong tonic, and it functions as an expectorant, diuretic, and immune regulator [2]; it also exists in the commercial market as a functional food [3]. For example, both ginseng under forest (planted in mountain forests and growing in natural environments) and *Panax quinquefolius* specimens can excite the central nervous system, reduce fatigue, and reduce blood sugar, which can be used to treat diabetes. The main pharmacological active component of ginseng is triterpene saponins (ginsenoside), which can be divided into dammarane tetracyclic triterpenes types and oleanane pentacyclic triterpenes types; dammarane tetracyclic triterpenes types are further divided into panaxadiol types (PPD) and panaxatriol types (PPT) [4]. Therefore, it is of great significance to use chemical methods to detect active metabolites, such as ginsenosides, sugars, and organic acids in ginseng in order to realize the quality identification of ginseng.

In recent years, mass spectrometry was widely used in the analysis of active metabolites in ginseng samples due to its advantages of high sensitivity, fast analysis speed, and less sample consumption. According to the principle of mass spectrometry, the identification of the characteristic components of different ginsenosides is the premise for the identification of ginsenoside metabolites [5,6,7]. Liquid chromatography (LC) and liquid chromatography-mass spectrometry (LC-MS) [5,8,9,10] are the main analytical techniques for ginsenosides detection and analysis, but these two methods require complex sample pretreatment that is time-consuming and laborious and may lead to loss of active components in the sample. Ginsenosides can also be detected with electrospray ionization mass spectrometry (ESI-MS) [11,12], but this method is not conducive to the direct analysis of complex matrices, such as ginseng, due to its strong background interference and low sensitivity. Electrospray desorption ionization (DESI) [13,14] and real-time direct analysis mass spectrometry (DART-MS) [15,16,17] have the advantages of no sample pretreatment and fast analysis speed and are used for the composition detection of ginseng samples. However, these two methods are limited to the analysis of a sample’s surface and do not detect its internal components. Therefore, it is very important to develop a rapid and sensitive internal information acquisition method without sample pretreatment. Recently, Zhang et al. developed an internal extractive electrospray ionization mass spectrometry (iEESI-MS) method [18], and Xu et al. used iEESI-MS to analyze six β-agonists in pork tissue samples [19]. Zhang et al. used iEESI-MS to successfully detect amino acids and sugars in garlic, strawberry, and other plant samples [20,21]. This method can directly obtain the internal information of tissue samples without sample pretreatment, hence avoiding the possible loss of component information and realizing the advantages of fast and real-time analysis. The method was applied in food science, plant metabolite analysis, and other fields [22].

Ginseng under forest and *Panax quinquefolius* specimens are excellent ginseng types that contain rich precious nutrients. In order to quickly and comprehensively identify their species and metabolic components, first, three ginseng samples were extracted with four different polar solvents using iEESI-MS sequentially, and the specific carbohydrate and organic acid metabolites of each sample were identified. To further characterize and visualize the differences between two samples of *Panax quinquefolius* of different origins (Canada and Jilin), an OPLS-DA model was established to demonstrate differences in the metabolic components between the two samples. Then, we selected a solvent to use iEESI-MS to detect 32 ginsenoside metabolites in ginseng under forest samples and two *Panax quinquefolius* samples. A total of 44 metabolites were identified in this study, obtaining richer research results than before. Moreover, this method can directly obtain the type and abundance information of the metabolites in ginseng samples with only one sampling in order to realize the differentiation of different ginseng varieties and to provide a new idea for the quality identification and efficacy evaluation of ginseng.

## 2. Materials and Methods

### 2.1. Materials and Reagents

Ginseng under forest samples were provided by Changchun University of Traditional Chinese Medicine. *Panax quinquefolius* samples from different origins (Quebec, Canada and Jilin, China) were purchased from pharmacies.

Three ginsenoside standards samples, namely, ginsenoside Rb1, ginsenoside Re, and ginsenoside Ro, were provided by Shanghai Yuanye Biotechnology Co., Ltd. (Shanghai, China, HPLC purity > 98%). Deionized water (18.2 MΩ/cm) was prepared using a Milli-Q Water Purification System (Billerica, MA, USA). HPLC-grade methanol was purchased from ROE Scientific Inc. (Newark, DE, USA). AR grade NH_4_Cl and NH_4_Ac were purchased from Xilong Chemical Co., Ltd. (Xilong, Guangxi, China). HPLC grade C_2_H_5_OH, HCOOH, and CH_3_CN were purchased from Anpel Laboratory Technologies (Shanghai) Inc. (Shanghai, China).

Ginsenoside standard samples were dissolved in methanol to obtain standard stock solvents of all three target analytes at a final concentration of 100 μg/mL. The solvents were stored at 4 °C prior to the iEESI-MS analysis.

### 2.2. Experimental Methods

#### 2.2.1. Mass Spectrometry Conditions

Appendix A shows the fingerprint spectra of ginsenoside standards samples in the negative ion mode and the positive ion mode. More ginsenoside signals can be detected in the negative ion mode and combined with literature analysis [23], the linear ion trap mass spectrum was detected in negative ion mode with a range of *m*/*z* 50~2000, the distance between the front of the ion source and the mass spectrometer inlet was about 3 mm, the capillary temperature was 250 °C, the capillary cone voltage was −23 V and the lens voltage was −228.11 V. The maximum ion entry time was 300 ms when performing tandem mass spectrometry analysis. The isolation window width of the parent ion was set to *m*/*z* 2.0, the activation value Q was 0.25, the collision activation time was 30 ms and the collision gas was helium (99.999% purity). Other parameters were optimized automatically with the LTQ-Tune software system in order to optimize the signal intensity of the target ions.

#### 2.2.2. Sample Pretreatment

After thawing, the ginseng samples were moistened with deionized water, and the surface was blotted with dust-free paper. In order to standardize the sample analysis procedure, 1 mm-thick circular slices were cut horizontally from the samples, and tissue blocks of the same size (about 1 × 1 × 1 mm^3^) were cut from the same radius of the circular slices and loaded into the sample chamber of the iEESI source for mass spectrometry detection.

#### 2.2.3. Method Optimization

Three representative ginsenoside (Rb1, Re, Ro) solvents were selected for iEESI-MS condition optimization analysis. The flow rate of the solvent was optimized between 5 and 20 μL/min, the ionization voltage was optimized between 2.0 and 5.5 kV, the distance between the front of the ion source and the mass spectrometer inlet was about 3 mm, and the capillary temperature was optimized between 100 and 350 °C. In this study, the effects of different solvents (0.5 mM ammonium chloride in methanol, 0.1% formic acid in water/ethanol (*v*:*v* = 1:1), 10 mM ammonium acetate in acetonitrile/methanol (*v*:*v* = 1:1), and 0.1% formic acid in acetonitrile/ethanol (*v*:*v* = 1:1)) on the signal intensity of ginsenoside metabolites were investigated.

## 3. Results and Discussion

Figure 1 shows the iEESI-MS apparatus diagram for the sequential analysis of the ginseng samples. Under the action of a pump, the solvent entered the ginseng sample through the capillary to extract the active components. The extraction solvent formed a conical droplet at the tip of the iEESI source under an electric field, which gave rise to gas-phase ions similar to those found in ESI, and entered into the entrance of the mass spectrometer to be detected. In this study, the ginsenoside metabolites in the ginseng under forest and *Panax quinquefolius* samples were detected via iEESI-MS, with 0.5 mM ammonium chloride in methanol solvent as the extraction solvent.

### 3.1. Optimization of the iEESI-MS

The effects of different extraction solvents (0.5 mM ammonium chloride in methanol, 0.1% formic acid in water/ethanol (*v*:*v* = 1:1), 10 mM ammonium acetate in acetonitrile/methanol (*v*:*v* = 1:1), and 0.1% formic acid in acetonitrile/ethanol (*v*:*v* = 1:1)) on the signal intensity of ginsenosides were investigated. As shown in Appendix A, the results showed that the methanol solvent with 0.5 mM ammonium chloride had the highest signal intensity of the target ions. Therefore, a methanol solvent containing 0.5 mM ammonium chloride was selected as the extraction solvent for the detection of ginsenoside metabolites in the samples in subsequent experiments. The flow rate of the extraction solvent was also one important factor that affected the extraction efficiency. When the flow rate was 15 μL/min, the signal intensity of the target ions was the highest. Other parameters, including the capillary voltage and capillary temperature, were also optimized to obtain the maximal intensity of three saponin ions. After optimization, the capillary voltage was −5 kV and the capillary temperature was 250 °C (Appendix A).

Figure 2 shows the mass spectrum fingerprints of the ginseng under forest, *Panax quinquefolium* (Canada), and *Panax quinquefolium* (Jilin) samples under the detection solvent analyzed with iEESI-MS. Dominant mass peaks, including *m*/*z* 341, *m*/*z* 377, *m*/*z* 719, and *m*/*z* 1061, were found in the mass spectra obtained from all the ginseng samples. In Figure 2a, the characteristic mass peak *m*/*z* 539 was present, but there was no mass peak *m*/*z* 1193 [24], as shown in Figure 2b,c. Thus, the chemical fingerprints of the three different ginseng samples were similar to each other, but diversity could also be recognized in some metabolites.

### 3.2. Analysis of Ginseng Samples with iEESI-MS for Single-Solvent and Sequential iEESI-MS

#### 3.2.1. Sequential Detection of Sugar and Organic Acid Metabolites in Ginseng under Forest Samples and *Panax quinquefolius* Samples

In addition to the methanol (0.5 mM ammonium chloride) solvent, the determination of sugar and organic acid metabolites in samples was also performed with other solvents using iEESI-MS. As shown in Figure 3a,c,e,g, the mass spectrum fingerprints of the ginseng under forest samples were obtained by placing the samples into the sample chamber of the iEESI and detecting them with 0.5 mM ammonium chloride in methanol, 0.1% formic acid in water/ethanol (*v*:*v* = 1:1), 10 mM ammonium acetate in acetonitrile/methanol (*v*:*v* = 1:1), and 0.1% formic acid in acetonitrile/ethanol (*v*:*v* = 1:1). Furthermore, the polarities of the above four solvents from strong to weak were displayed by 0.5 mM ammonium chloride in methanol, 0.1% formic acid in water/ethanol (*v*:*v* = 1:1), 10 mM ammonium acetate in acetonitrile/methanol (*v*:*v* = 1:1), and 0.1% formic acid in acetonitrile/ethanol (*v*:*v* = 1:1) [22,25,26,27,28,29]. In this order, the same sample in the sample chamber was detected sequentially. The obtained mass spectrum fingerprints of the ginseng under forest samples are shown in Figure 3b,d,f,h.

The results of the four solvent sequence detection for the ginseng under forest samples were compared with the results of the separate detection of ginseng under forest samples. The mass spectrum of the first solvent (0.5 mM ammonium chloride in methanol) was similar. When the second solvent (0.1% formic acid in water/ethanol (*v*:*v* = 1:1)) was used to detect the sample, the signals of the peaks *m*/*z* 191 and *m*/*z* 133 in the mass spectrum obtained using the sequential detection of samples were stronger than those detected using 0.1% formic acid water/ethanol (*v*:*v* = 1:1) alone, presumably due to the effect of the first solvent (0.5 mM ammonium chloride in methanol), which enhanced the signal of active metabolites in the ginseng samples [30]. When the third solvent (10 mM ammonium acetate in acetonitrile/methanol (*v*:*v* = 1:1)) was used to detect the sample, the peaks of *m*/*z* 119, *m*/*z* 255, and *m*/*z* 1193 in the mass spectrum obtained via separate detection of the solvent had almost no signal or low signal; however, the relative signal strength of the solvent in the sequence detection was strong. It is speculated that this was due to the residue of the second solvent (0.1% formic acid in water/ethanol (*v*:*v* = 1:1)). The peak intensity of the mass spectrum obtained via the sequential detection of 0.1% formic acid in acetonitrile/ethanol (*v*:*v* = 1:1) solvent for the fourth solvent was obviously lower than that obtained via the separate detection of the solvent. It is speculated that the components in the sample were extracted in the first three solvents. It can be seen that sequential extraction with iEESI-MS was able to obtain more complete sample information.

Based on the relevant literature references, sequential iEESI-MS was also realized for the detection of active metabolite components in three ginseng samples, whose results are shown in Table 1. In this table, a, b, c, and d indicate the results of the four solvents on the samples, i.e., the 0.5 mM ammonium chloride in methanol, 0.1% formic acid in water/ethanol (*v*:*v* = 1:1), 10 mM ammonium acetate in acetonitrile/methanol (*v*:*v* = 1:1), and 0.1% formic acid in acetonitrile/ethanol (*v*:*v* = 1:1) solvents, respectively, for the determination of the metabolite components of ginseng under forest, *Panax quinquefolius* (Canada), and *Panax quinquefolius* (Jilin) samples via iEESI-MS. According to the data in Table 1, the methanol (0.5 mM ammonium chloride) solvent showed the best extraction effect for the 12 active components, with sucrose extracted the best under different solvents for three samples. Among them, methyl gallate 3-0-β-D-glucoside and isoconiferoside were only detected in the ginseng under forest samples. The active metabolite components detected were similar for both types of *Panax quinquefolius*, but oleic acid was only detected in the methanol (0.5 mM ammonium chloride) solvent for the *Panax quinquefolius* sample (Canada). The detection of sugar and organic acid metabolites could also be used for the identification of ginseng quality.

Figure 4 shows the MS^2^ spectra of the active metabolite components measured in the samples. In the MS^2^ spectrum of *m*/*z* 117→, the fragment ions at *m*/*z* 99 and 73 were observed (Figure 4a), which corresponded to [M-H-H_2_O]^−^ and [M-H-CO_2_]^−^, respectively. Therefore, *m*/*z* 117 was assigned to succinic acid. In the MS^2^ spectrum of *m*/*z* 133→, the fragment ions at *m*/*z* 115 and 97 were observed (Figure 4b), corresponding to [M-H-H_2_O]^−^ and [M-H-2H_2_O]^−^, respectively. Therefore, *m*/*z* 133 was assigned to malic acid. In the MS^2^ spectrum of *m*/*z* 179→, the fragment ions at *m*/*z* 161, 143, and 119 were observed (Figure 4c), corresponding to [M-H-H_2_O]^−^, [M-H-2H_2_O]^−^, and [M-H-C_2_H_4_O_2_]^−^, respectively. Therefore, *m*/*z* 179 was assigned to fructose. In the MS^2^ spectrum of *m*/*z* 191→, the fragment ions at *m*/*z* 173 and 111 were observed (Figure 4d), corresponding to [M-H-H_2_O]^−^ and [M-H-2H_2_O-CO_2_]^−^, respectively. Therefore, *m*/*z* 191 was assigned to citric acid. In the MS^2^ spectrum of *m*/*z* 503→, the fragment ions at *m*/*z* 341, 323, and 179 were observed (Figure 4e), corresponding to [M-H-Glc]^−^, [M-H-H_2_O-Glc]^-^, and [M-H-2Glc]^−^, respectively, or corresponding to [M-H-Ara-CH_2_O]^−^, [M-H-H_2_O-Glc]^−^, and [M-H-2Glc]^−^, respectively, Therefore, *m*/*z* 503 was assigned to raffinose or isoconiferoside. The addition of ammonium chloride to the solvent methanol (0.5 mM ammonium chloride) also led to the formation of a chlorination peak for some of the active components. The peak at *m*/*z* 215 corresponded to the chloride adduct; the loss of HCl led to the deprotonated molecule of fructose *m*/*z* 179. The chloride anions were produced via the electrochemical reduction of chlorinated solvents at the ESI capillary, as reduction processes were inherent to the negative ion ESI process, leading to the formation of chloride adducts [M+Cl]^−^, which competed with the formation of the deprotonated molecules [M-H]^−^. The relationship between the two species was a difference of 36 Da [24], as shown in Figure 4f.

#### 3.2.2. Analysis of Ginsenoside Metabolites in Ginseng under Forest and *Panax quinquefolius* Specimens

Under optimal conditions, the ginsenosides in ginseng under forest samples and two *Panax quinquefolius* samples were determined with 0.5 mM ammonium chloride in the methanol solvent. The test results are shown in Table 2. A total of 32 ginsenoside metabolites (for the structures, see Appendix A) were identified, including 11 panaxadiol (compounds **1** to **11**), 14 panaxatriol (compounds **12** to **25**), and 3 oleanane types (compounds **26** to **28**); in addition, 4 ginsenoside derivatives (compounds **29** to **32**) were also identified. In the negative ion mode, the major ions of most ginsenosides were [M-H]^−^. Figure 5 shows the MS^2^ spectrum of typical ginsenosides detected via iEESI-MS under methanol (0.5 mM ammonium chloride). For example, the [M-H]^−^ of compounds **1** and **14** were both *m*/*z* 783 ions, and compound **14** was a panaxatriol saponin; in the MS^2^ spectrum of *m*/*z* 783→, the fragment ions at *m*/*z* 637, 619, and 475, were observed (Figure 5b), corresponding to [M-H-Rha]^−^, [M-H-Glc]^−^, and [M-H-Rha-Glc]^−^, respectively. Compound **1** was a panaxadiol saponin; in the MS^2^ spectrum of *m*/*z* 783→, the fragment ions at *m*/*z* 621 and 459 were observed (Figure 5b), corresponding to [M-H-Glc]^−^ and [M-H-2Glc]^-^, respectively. Therefore, in the identification of ginsenosides, information about the glycosyl substituents could be obtained based on neutral loss, and the conventional cleavage modes and pathways of ginsenosides that are applicable to most ginsenosides are shown in Appendix A. The fragment ions formed after the cleavage of panaxadiol-type saponin paralogues were *m*/*z* 945, *m*/*z* 783, *m*/*z* 621, and *m*/*z* 459 in most cases; *m*/*z* 637 and *m*/*z* 475 were generally formed by panaxatriol type saponin paralogue cleavage; and *m*/*z* 455 was usually formed by oleanane-type ginsenoside paralogues. The neutral loss of 132, 146, and 162 Da may have indicated the elimination of a pentose group, rhamnose group, and glucose group, respectively, and the neutral loss of 294, 308, 324, and 486 Da may have indicated the elimination of glucose-pentose, glucose-rhamnose, two molecules of glucose, and three molecules of glucose, respectively.

Based on the above interpretation and the relevant literature references, the MS^2^ spectra of ginsenoside metabolites shown in Figure 5, Figure 5b,e–h are the MS^2^ spectra of panaxadiol saponins; Figure 5a,b,d,e are the MS^2^ spectra of panaxatriol saponins; and Figure 5c is the MS^2^ spectrum of oleanane-type saponins. According to the mass spectrum peaks in Figure 5b,e, and considering the [M-H]^−^ in Table 2, they were identified as both ginseng panaxadiol saponins Rg3 and Rd and ginseng panaxatriol saponins Rg2 and Re.

Table 2 lists the ginsenosides in the three ginseng samples detected with iEESI-MS, among which Rg3, Rd, Rc, Rb1, Rb2, Rb3, quinquenoside-R1, R0, Korean-R2, notoginsenoside-A, Re, and V were the common ginsenoside metabolites in the three ginseng genus samples detected using iEESI-MS. Furthermore, the quality of the three ginseng samples could be identified according to the characteristics of ginsenoside metabolites detected in the three samples. Notoginsenoside-R2, Rs1, and Rs2 were the characteristic ginsenoside metabolites in the ginseng under forest samples; pseudo-Rc1 was the characteristic ginsenoside metabolite in the *Panax quinquefolius* sample (Canada); Rs3, Rg2, Ma-Rg1, Ma-Rf, Ma-(20-glu-Rf), Ma-notoginsenoside-N, Ma-Re1, Ma-Re2, and Ma-Re3 were the characteristic ginsenoside metabolites in the *Panax quinquefolius* sample (Jilin). Based on a comparison of the iEESI-MS detection results of the ginsenoside metabolites in the three samples, the three ginseng samples could be identified.

The addition of ammonium chloride in the methanol solution (0.5 mM ammonium chloride) enhanced the signals of ginsenosides, and formed some of the chlorination peaks of the ginsenosides, as shown in Appendix A for the MS^2^ spectrum of the Rg1/Rf chlorination peak *m*/*z* 835 and Re/Rd chlorination peak *m*/*z* 981; in the negative ion mode, ginsenosides [M+Cl]^−^ and [M-H]^−^ had a difference of 36 Da, forming *m*/*z* 799 and *m*/*z* 945, respectively [24].

### 3.3. Multivariate Statistical Analysis of Panax quinquefolius Specimens from Different Origins

To further characterize and visualize the differences between two samples of *Panax quinquefolius* of different origins (Canada and Jilin), the global iEESI-MS fingerprint data of the sequential detection of *Panax quinquefolius* samples under 0.5 mM ammonium chloride in methanol, 0.1% formic acid in water/ethanol (*v*:*v* = 1:1), 10 mM ammonium acetate in acetonitrile/methanol (*v*:*v* = 1:1), and 0.1% formic acid in acetonitrile/ethanol (*v*:*v* = 1:1) solvents were obtained. The obtained data of the signal peaks, sample species, and ionic intensities were analyzed using multivariate statistics with SIMCA-p software. Figure 6a,b show the scatter plots of the OPLS-DA scores for *Panax quinquefolius* samples with different origins. It can be found that two *Panax quinquefolius* samples showed significant differences, indicating that there were clear differences in the chemical compositions of *Panax quinquefolius* samples from different origins. The results showed that the parameters R2Y and Q2 of the newly developed OPLS-DA model were 0.991 and 0.691, respectively, indicating that the model exhibited good predictive ability and could be used to distinguish ginseng samples of different origins.

Variable importance in the projection (VIP) can be used to measure the impact and interpretive ability of each metabolite’s expression pattern on the classification discrimination of each group. To identify the differentiated metabolites that contributed the most to the group separations, the feature selections were performed using a scatter plot derived from the constructed OPLS-DA. The VIP plots of the differential metabolic chemical markers of *Panax quinquefolius* samples are shown in Figure 6c. The chemical identities of these markers are listed in Table 1 and Table 2. A total of 26 MS signals from *Panax quinquefolius* samples were screened by taking a VIP score > 1.0 as the evaluation standard, including ginsenoside metabolites (e.g., *m*/*z* 931, *m*/*z* 955, *m*/*z* 825, *m*/*z* 769, *m*/*z* 987, *m*/*z* 637, *m*/*z* 1047, *m*/*z* 885, *m*/*z* 799, and *m*/*z* 1077), and MS signals at *m*/*z* 673, *m*/*z* 1083, *m*/*z* 921, *m*/*z* 861, *m*/*z* 835, and *m*/*z* 1023 were assigned to chlorination peaks of ginsenosides [Rh1+Cl]^−^, [Ma-(20-glu-Rf)+Cl]^−^ or [Ma-(notoginsenoside-N)+Cl]^−^ or [Ma-(Re1/Re2/Re3)+Cl]^−^, [Ma-(Rg1)+Cl]^−^ or [Ma-(Rf)+Cl]^−^, [Rs3+Cl]^−^, [Rg1+Cl]^−^ or [Rf+Cl]^−^, and [(pseudoginsenoside-Rc1)+Cl]^−^, respectively. Furthermore, other active metabolic components, namely, *m*/*z* 345, *m*/*z* 503, *m*/*z* 179, *m*/*z* 255, *m*/*z* 341, and MS signals at *m*/*z* 539, *m*/*z* 215, and *m*/*z* 317, were assigned to chlorination peaks of active metabolites [raffinose+Cl]^−^, [fructose+Cl]^−^, and [oleic acid+Cl]^−^, and the MS signals at *m*/*z* 719 and *m*/*z* 1061 were assigned to the chlorination peaks of compounds [2sucrose+Cl]^−^ and [3sucrose+Cl]^−^, respectively. The differences in these metabolite components in *Panax quinquefolius* samples could be used as key characteristic indicators for the origin identification of *Panax quinquefolius* samples. Therefore, we concluded that the environmental growth factors caused significant differences in the chemical qualities of *Panax quinquefolius* samples.

## 4. Conclusions

In this study, we used an iEESI-MS method to identify the metabolites in the ginseng under forest and *Panax quinquefolius* samples without tedious pretreatment in order to obtain information on the internal composition of the samples. A total of 44 metabolites, including ginsenosides, sugars, and organic acids, were detected and identified in ginseng samples based on tandem MS information, experiments with standard compounds, and earlier reports. The OPLS-DA analysis of two kinds of *Panax quinquefolius* samples from different origins showed obvious separation. Compared with other ionization methods (ESI, DESI, etc.), this iEESI ionization method did not require gas assistance and was not limited to the detection of the surface layer of the sample, but could extract the internal chemical components of the tissue sample. iEESI-MS has the advantages of fast detection, real-time analysis of data, and comprehensive information acquisition. In the future, metabolite fingerprinting studies based on iEESI-MS methods will be more widely applied to the screening and characterization of traditional Chinese medicines, which will help to understand their activity differences and quality identification at the molecular level.

## Figures and Tables

**Figure 1 foods-12-01152-f001:**
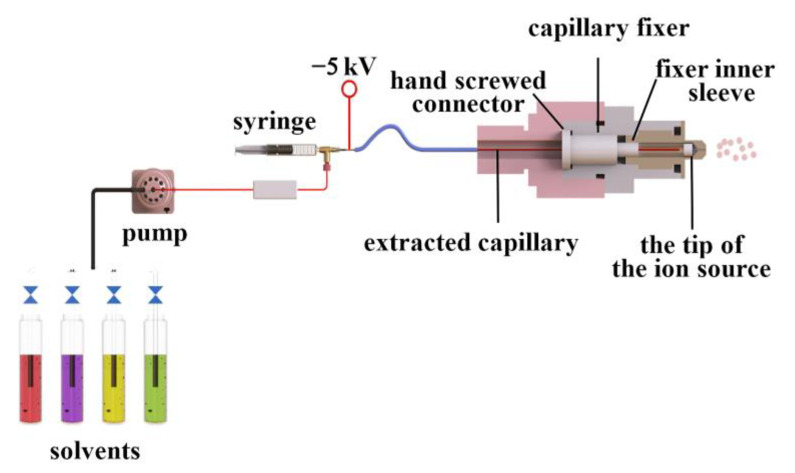
The apparatus diagram of the iEESI-MS.

**Figure 2 foods-12-01152-f002:**
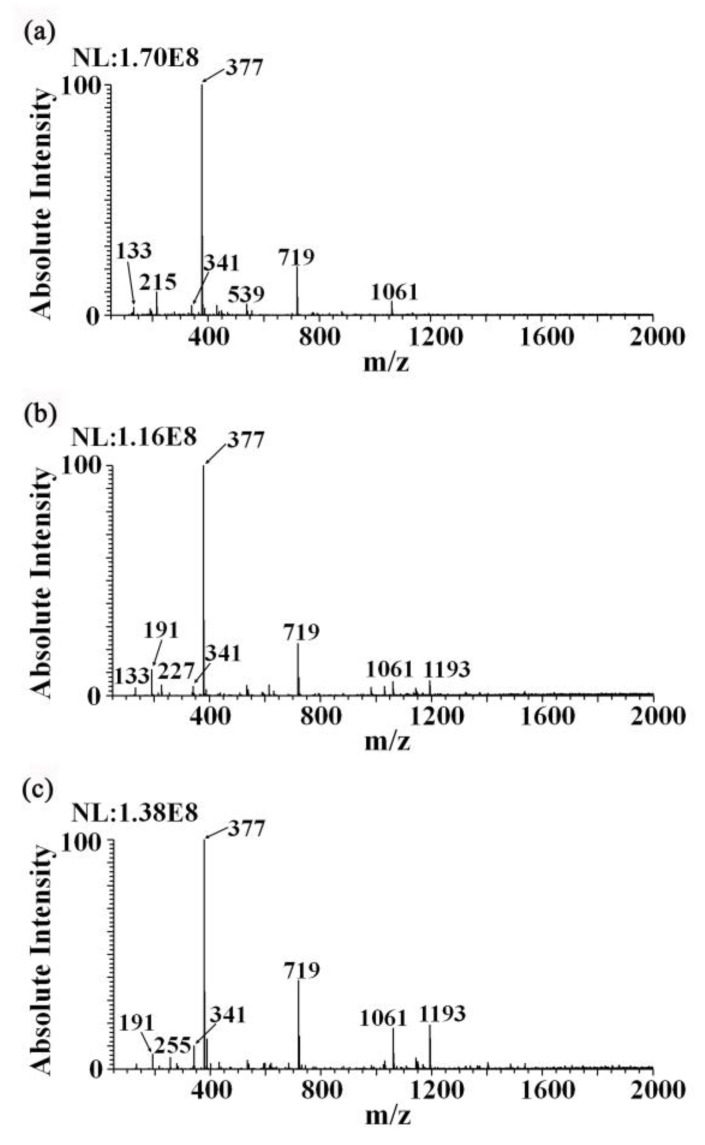
The fingerprint data of methanol (0.5 mM ammonium chloride) extracted from three samples detected using iEESI-MS: (**a**) fingerprint data of the ginseng under forest sample, (**b**) fingerprint data of the *Panax quinquefolium* (Canada) sample, and (**c**) fingerprint data of the *Panax quinquefolium* (Jilin) sample.

**Figure 3 foods-12-01152-f003:**
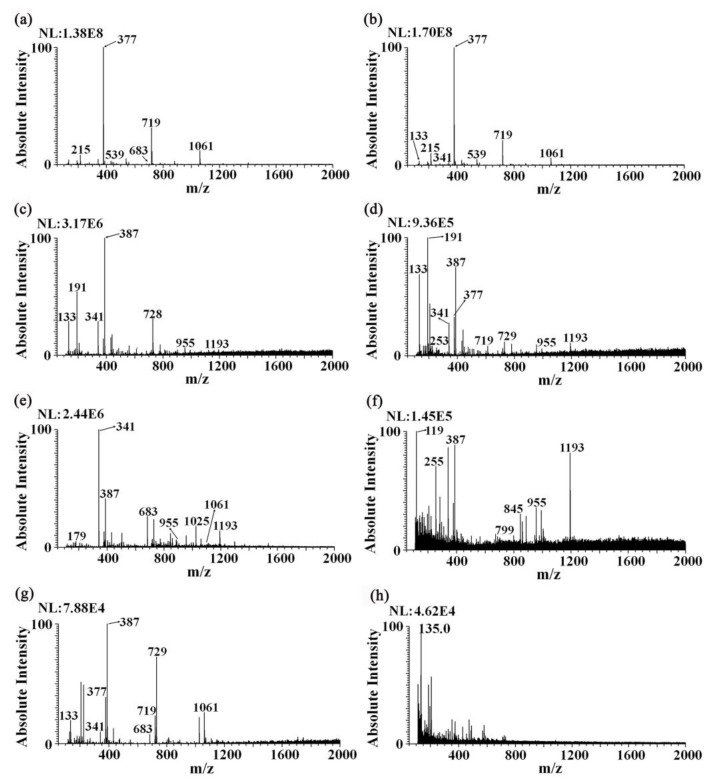
The fingerprint spectra of different solvents for the detection of ginseng under forest samples detected with iEESI-MS. (**a**,**c**,**e**,**g**) in the left column correspond to the fingerprint spectra of the ginseng under forest samples under 0.5 mM ammonium chloride in methanol, 0.1% formic acid in water/ethanol (*v*:*v* = 1:1), 10 mM ammonium acetate in acetonitrile/methanol (*v*:*v* = 1:1), and 0.1% formic acid in acetonitrile/ethanol (*v*:*v* = 1:1) solvents, respectively; (**b**,**d**,**f**,**h**) in the right column correspond to the fingerprint spectra of the ginseng under forest samples extracted sequentially in 0.5 mM ammonium chloride in methanol, 0.1% formic acid in water/ethanol (*v*:*v* = 1:1), 10 mM ammonium acetate in acetonitrile/methanol (*v*:*v* = 1:1), and 0.1% formic acid in acetonitrile/ethanol (*v*:*v* = 1:1) solvents, respectively.

**Figure 4 foods-12-01152-f004:**
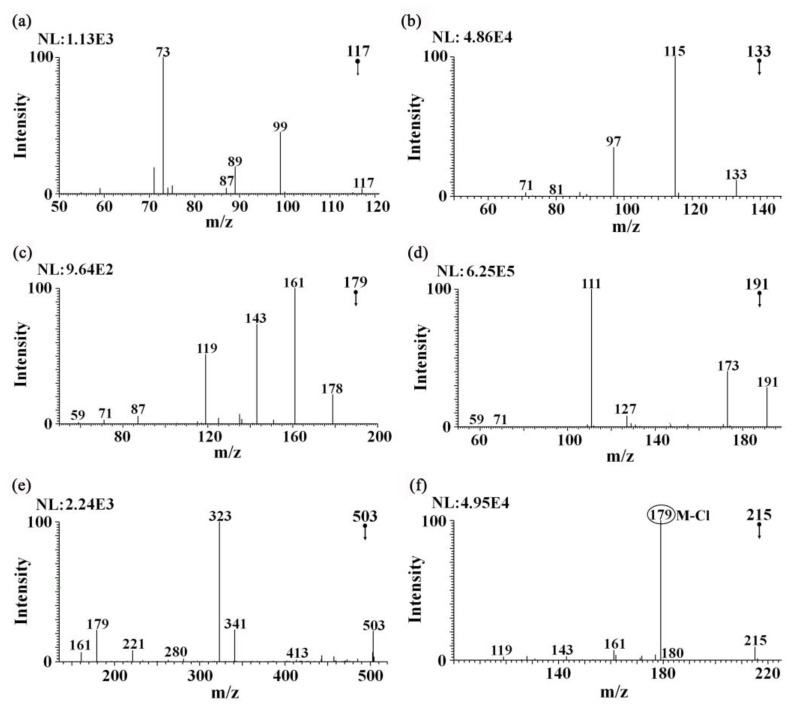
Tandem MS analysis of characteristic ions of the active components in the ginseng under forest and two *Panax quinquefolium* samples detected with iEESI-MS: (**a**) MS^2^ spectrum of *m*/*z* 117→, (**b**) MS^2^ spectrum of *m*/*z* 133→, (**c**) MS^2^ spectrum of *m*/*z* 179→, (**d**) MS^2^ spectrum of *m*/*z* 191→, (**e**) MS^2^ spectrum of *m*/*z* 503→, and (**f**) MS^2^ spectrum of *m*/*z* 215→.

**Figure 5 foods-12-01152-f005:**
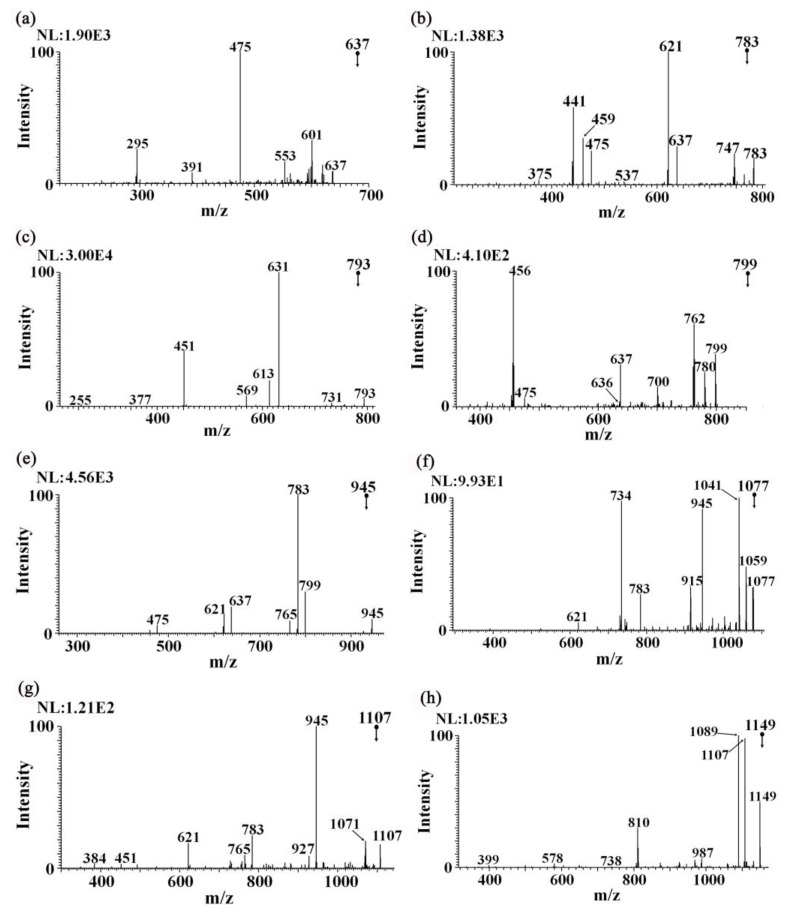
Tandem MS analysis of typical ginsenosides in ginseng under forest and two *Panax quinquefolium* samples detected with iEESI-MS: (**a**) MS^2^ spectrum of *m*/*z* 637→, (**b**) MS^2^ spectrum of *m*/*z* 783→, (**c**) MS^2^ spectrum of *m*/*z* 793→, (**d**) MS^2^ spectrum of *m*/*z* 799→, (**e**) MS^2^ spectrum of *m*/*z* 945→, (**f**) MS^2^ spectrum of *m*/*z* 1077→, (**g**) MS^2^ spectrum of *m*/*z* 1107→, and (**h**) MS^2^ spectrum of *m*/*z* 1149→.

**Figure 6 foods-12-01152-f006:**
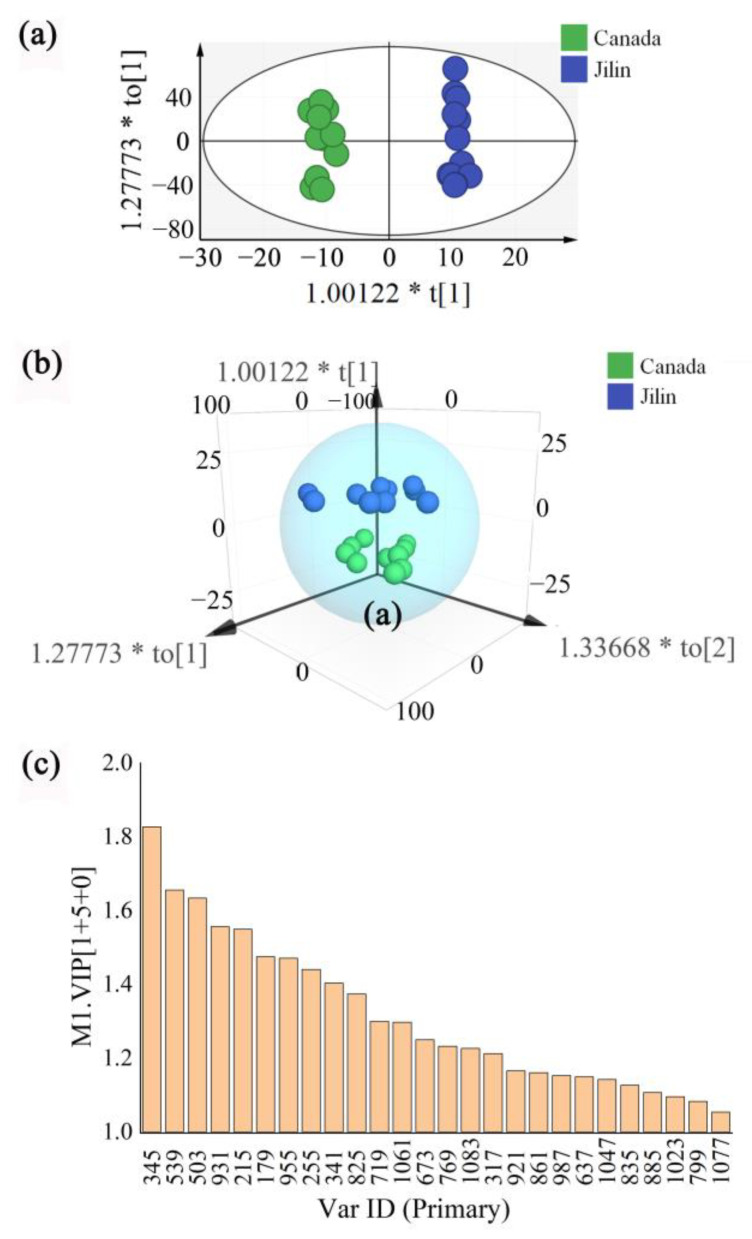
Scatter plot of OPLS-DA scores of two *Panax quinquefolius* samples detected using sequential iEESI-MS: (**a**) score scatter plot of the 2D OPLS-DA model, (**b**) score scatter plot of the 3D OPLS-DA model, and (**c**) bar plot with the OPLS-DA model of the VIP.

**Table 1 foods-12-01152-t001:** Distribution of active metabolite components in the ginseng and *Panax quinquefolius* samples.

Identity	[M-H]^−^	Ginseng under Forest	Canada	Jilin	Fragment Ion	Reference
a	b	c	d	a	b	c	d	a	b	c	d
Succinic acid	117	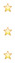							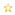				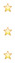	99, 89, 73	[31]
Malic acid	133	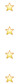	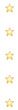		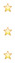	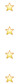	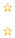	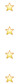	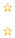	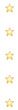	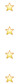	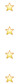	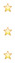	115, 87, 71	[31]
Fructose	179	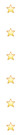		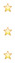		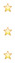		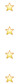				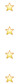		161	[31]
Citric acid	191	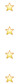	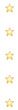			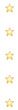	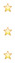		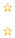	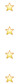	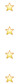	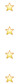	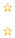	173, 129, 111	[31]
Palmitoleic acid	253		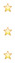											235, 209, 193	[32]
Palmitic acid	255	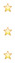		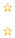		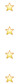	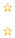	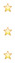		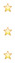		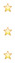		237, 211, 184	[32]
Oleic acid	281					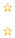								263, 245, 237	[32]
α-Maltose	341	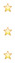	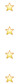	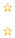	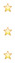	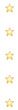	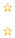	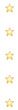	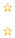	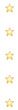	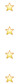	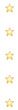	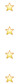	179, 149	[33]
Sucrose	341	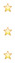	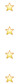	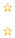	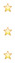	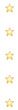	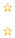	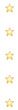	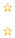	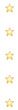	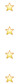	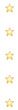	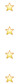	215, 179, 161	[34]
Methyl gallate3-O-β-D-glucoside	345	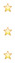												309, 183	[35]
Raffinose	503	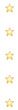		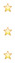		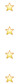								341, 323, 179, 161	[31]
Isoconiferoside	503			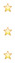										341	[35]

Note: In the table, a, b, c, and d refer to 0.5 mM ammonium chloride in methanol, 0.1% formic acid in water/ethanol (*v*:*v* = 1:1), 10 mM ammonium acetate in acetonitrile/methanol (*v*:*v* = 1:1), and 0.1% formic acid in acetonitrile/ethanol (*v*:*v* = 1:1), respectively. 
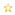
 symbols are used to represent the signal strength, where the more 
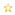
 symbols there are, the better the signal. 
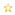
 represents a signal strength of 10^1^–10^2^, 
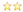
 represents a signal strength of 10^2^–10^3^, 
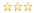
 represents a signal strength of 10^3^–10^4^, 
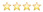
 represents a signal strength of 10^4^–10^5^, 
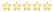
 represents a signal strength of 10^5^–10^6^, and 
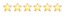
 represents a signal strength of 10^6^–10^7^.

**Table 2 foods-12-01152-t002:** Distribution of ginsenosides in the three ginseng samples.

No.	[M-H]^−^	Ginsenoside	Ginseng under Forest	Canada	Jilin	Fragment Ion	References
1	783	Rg3	√	√	√	621, 459	[36]
2	825	Rs3			√	765, 663	[37]
3	945	Rd	√	√	√	799, 783, 765, 637, 621, 475, 459, 369	[5]
4	987	Pseudoginsenoside-Rc1		√		945, 783	[38]
5	1077	Rc	√	√	√	945, 915, 783, 621	[5]
6	1077	Rb2	√	√	√	945, 784, 783, 621, 459	[5]
7	1077	Rb3	√	√	√	945, 784, 783, 621	[5]
8	1107	Rb1	√	√	√	945, 825, 783, 621, 459	[5]
9	1119	Rs1	√			1077, 945, 928, 783	[37]
10	1119	Rs2	√			1077, 945, 928, 783	[37]
11	1149	Quinquenoside-R1	√	√	√	1107, 1089, 987	[35]
12	637	Rh1	√		√	553, 475, 391, 294	[39]
13	769	Notoginsenoside-R2	√			730, 638, 637, 619, 475	[5]
14	783	Rg2			√	637, 619, 475	[36]
15	799	Rg1	√		√	637, 475	[36]
16	799	Rf	√		√	637, 475, 323	[5]
17	885	Ma-Rg1			√	841, 781	[5]
18	885	Ma-Rf			√	841, 781	[5]
19	931	Notoginsenoside-R1	√		√	799, 637, 619, 475	[37]
20	945	Re	√	√	√	799, 783, 765, 637, 475, 474	[5]
21	1047	Ma-(20-glu-Rf)			√	1003, 961	[5]
22	1047	Ma-Notoginsenoside-N			√	1003, 961	[3]
23	1047	Ma-Re1			√	1003, 961	[3]
24	1047	Ma-Re2			√	1003, 961	[3]
25	1047	Ma-Re3			√	1003, 961	[3]
26	793	Chikusetsusaponin-Iva		√	√	631, 569, 455	[5]
27	793	Zingibroside-R1		√	√	569, 455	[5]
28	955	R0	√	√	√	793, 731, 613, 523	[5]
29	955	Notoginsenoside-R3	√	√		799, 637, 475	[40]
30	1123	Korean-R2	√	√	√	961, 799, 637, 476	[41]
31	1123	V	√	√	√	961, 799, 637, 476	[42]
32	1123	Notoginsenoside-A	√	√	√	961, 799, 637, 476	[40]

Note: Ma, malonyl.

## Data Availability

Acquired data are available upon request.

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
