# Peer review of "Metabolite Fingerprinting for Identification of Panax ginseng Metabolites Using Internal Extractive Electrospray Ionization Mass Spectrometry"

_foods, 2023, doi:10.3390/foods12061152_

Round 1

Reviewer 1 Report

Concept and idea is good but the overall presentation, experimental design and clarity is very much poor. Introduction is not clear. Even the authors have not cleared what was the difference between samples of ginseng. What was the actual difference on which the design of the experiment was based upon. Also they have not cleared what was the actual form of the samples. English is very poor with lot of grammatical mistakes.  My suggestions are as follows;

1. Please re-write the manuscript with clear understanding and assistance may be sought from some expert in English.

2. Clarify the experiment design.

3. Clear that what was the base of experimental design?

4. Clearly define the objectives of the study.

5. Link your conclusions with the objectives.

6. Though there is information about the sample types in material and methods but nothing such present in introduction where the authors have to define the objective of the study.

Author Response

请参阅附件。

Reviewer 2 Report

Dear authors, your manuscript entitled "Metabolomic approach for identification of Panax ginseng metabolites using internal extractive electrospray ionization mass spectrometry", describes an useful methodology for a rapid screen of ginseng extracts. However, some improvements should be considered. 

Please, below are my comments.

Pg. 4 – Lines 142 – 147.

Comment: In the paragraph 2 of Pg 4 it was discussed the differentiation of the chemical profile of ginseng samples using methanol containing 0.5 mM ammonium chloride as eluting solvent.

Comparing the spectra profiles, it was concluded that spectrum 2a presents a characteristic mass peak at m/z 539, but not the peak m/z 1193 that it is only presents in spectra 2b and 2c. Based on this result it was concluded that it is possible differentiated the ginseng samples using methanol containing 0.5 mM ammonium chloride as eluting solvent.

Why peak m/z 539 is a characteristic mass peak of gingeng under florest sample? The peak also exists in spectra 2b-2c, and it seems to correspond to a chloride adduct.

The peak m/z 1193 wasn´t identified as a compound belonging to the chemical fingerprint of these samples. The three spectra are very similar, the differentiation of the chemical fingerprint probably was achieved on the lower signals present in the spectra. Please comment.

Pg. 4 – section 3.2.

Comment: In this section it was discussed the efficiency of extraction of the metabolites  using four  eluting solvent, and also a sequential elution performed with the four solvents. The discussion of the data  is confused, and it is not clear if for the small metabolites (phenolic acids and sugars) the final data result from a sequential extraction, and only the ginsenoside metabolites are extracted with methanol containing 0.5 mM ammonium chloride. I suggest to review and clarify the text of this section.

Pg. 6 - Table 1

Comment: The table lists the m/z values of the deprotonated molecules of the metabolites along with the respective fragment ions. In column 6 “Fragment ions”, the m/z values assigned to the precursor ions isolated in the ion trap must be removed. These are not fragment ions.

In the note of Table 1 it is not clear what one star or two stars mean, and so on. Does the number of stars represent the efficiency of the sequential extraction?

The precursor ion with m/z 683 was assigned to 2Sucrose. The m/z 683 value  corresponds to the gas phase dimer [2M-H]- formed during the ionization process. This comment also applies to the sucrose-3 chloride adduct mentioned on pg. 12 line 345. Considering that both species are gas-phase dimers/trimers, they should not be considered active metabolites.

Pg. 8 - Lines 233 - 235

…., the major ion pair of fructose [M+Cl]- and [M-H]- had a difference of 36 Da, losing m/z 36 (-Cl) to form its original fructose molecular ion peak at m/z  179,…

Comment: Under ESI conditions the species [M-H]- and [M+Cl]- aren´t ion pair ions. The chloride anions may be produced via electrochemical reduction of chlorinated solvents at the ESI capillary, as reduction processes are inherent to the negative ion ESI process, leading to the formation of chloride adducts [M+Cl]-, that´s competed with the formation of the deprotonate molecules [M-H]-.The relationship between the two species is a difference of 36 u.

The spectrum 4f presents a precursor ion m/z 215 without the characteristic isotopic distribution of the presence of a Cl atom in its ionic structure. If the peak at m/z 215 corresponds to the chloride adduct, the loss of HCl leads to the deprotonated molecule of fructose m/z 179.  

Please also correct the term “ion pair” along manuscript.

Pg. 8 – line 251

…..the excimer ion peaks [M-H]-.

Comment: The term excimer peak ions could not be applied to deprotonated molecules formed in a charged, nebulized solution of analytes. The term can be applied to species formed for instance by techniques as Deep-ultraviolet Laser Ablation Electrospray Ionization Mass Spectrometry.

Although,  the precursor ions isolated in the ion trap are excited by low collision induced dissociation with a neutral molecule, they could not be consider excimer species.

Please remove the expression.

Pg. 9 – Table 2

Please remove from the column “Fragment ion” the m/z values corresponding to the precursor ions.

Please check the fragment ions generated from the precursor ions of the ginsenoside metabolites listed in Table 2 together with their attribution discussed along lines 268 -295

Examples: (‘)

metabolite             precursor ion             fragment ion                 attribution

    8 and 9              m/z 1077                     m/z 784                 loss of 293 u (?)

                                                                   m/z 734                loss of 343 u  (?) *

 11 and 12             m/z 1119                    m/z 928                  loss of  149 u (?)

17    (spectrum 5d)   m/z 799                    m/z 762                  loss of 37 u (?)**

*the value does not corresponds to the loss of 2Glc as mentioned in lines 271 -272

 * not loss of 2H2O (line 280)

Pg- 10 – Fig 5  and Lines 271 -275 c

Spectra 5f and 5g are assigned to panaxadiol saponins based on the presence of fragment ions at m/z 945, 783 and 621.

The identification of the fragment ions with m/z 637 and 475 in MS2 spectra 5(a),(b), (d)  and (e) leads to the conclusion that these spectra can be attributed to panaxatriol saponins.

Comments:

Spectrum 5b contains  fragment ions with m/z 637 and 475, but intense peaks at m/z 621 and 453. In Table 2 the deprotonated molecules m/z 783 are assigned to Rg3 ginsenosides which are panaxadiol-type saponins, and also to one Rg2 , a panaxatriol-type saponins. Please review the text.

Spectrum 5c: along lines 284-286 it is analysed the fragmentation dissociation pattern of the precursor ion m/z 793.  No signal of one fragment ion with m/z 455 is observed in the MS2 spectrum. Consequently, the presence of a glucuronide unit is not identified. Based on this spectrum can the metabolite be assigned to an oleanane type saponin?

Spectrum 5h show the MS2 spectrum of a precursor ion with m/z 1149 assigned to a Quinquenoside R1, a panaxadiol with 4 Glc, one of them containing an Ac group.

Assignment is based on 3 fragment ions with m/z 1107 (loss of 42 u); 1089 (loss of 60 u) and 987 (loss of 162 u). In addition to these fragments, the spectrum also exhibits signals at m/z 810 (loss of 339 u), 578 (loss of 571 u) and 399 (loss of 570 u). The formation of these fragments is not easily explained from the ionic structure of the proposed panaxadiol ginsenoside.

The authors propose that the characteristic glycogenic ions of a Panaxadiol-type ginseng must include fragments at m/z 945, 783, 621, 459 (Doc SI – Fig S2). Analysing the signals of spectrum 5h no evidence of that characteristic ions where found. How do you explain that the deprotonated molecule with m/z 1149 really corresponds to the proposed metabolite?

General Comment:  The six spectra shown in Fig.5 are tandem mass spectra; its analysis supports the attribution/confirmation of the ionic structure of a proposed molecule. As mentioned along the manuscript and noted in Fig. S2 of the Supporting Information, the analysis of the MS2 spectra was based on the identification of fragment ions characteristic of each type of terpenoid. For example, spectrum 5e was assigned to a ginsenoside Re, a panaxatriol saponin type, based on the identification of the fragments m/z 637 and 475 considered characteristic of this saponin-type. But the spectrum also shows an intense peak at m/z 783 (loss of 162 u), a smaller one at m/z 621 (loss of 2x162 u) and a small fragment m/z 459 not marked on the spectrum. These three fragments are part of the list of characteristic fragments of the panaxadiol-saponin-type. What is the justification for identifying the metabolite as a panaxatriol and not as a panaxadiol?

On the other hand, spectrum 5e which displays both type of ions, the precursor ion m/z 783 was assigned to two distinct saponin-types (Rg3 and Rg2) metabolites.

Furthermore, if the 5g (m/z 1107) and 5e (m/z 945) spectra are compared, it is concluded that both present a very similar fragmentation profile. But the first one was attributed to a panaxadiol-type saponin, while the second one to a panaxatriol-type saponin. What was the criteria followed in the identification?

These comments clearly indicate that in the eluting solution the presence of isobaric precursor ions may occur.  Considering that the analysis does not involve chromatographic separation or accurate mass measurements, the attribution of isobaric ions must be based on good tandem mass spectra and a careful data analysis. The data interpretation should be reviewed.

 Supplementary Information

Figure S2

Comments:

Molecular ion                    should be               deprotonated molecules

m/z 637, 475                   correspond to  panaxatriol type

glucose                               should be               hexose

rhamnose                          should be               deoxyhexose

According with your scheme, the generated fragment ions can be glycogen and sugar ions. The presence of the last type of fragment ions is associated with characteristic mass losses,  for hexoses, losses of 180 and 162 u are observed. This means that these values must be assigned as neutral species and not as ionic species, as the authors mention in Pg. 9 line 259 of the manuscript. The scheme must be redrawn.

Fig. S3

The Figure shows two tandem mass spectra of two chloride adduct of ginsenosides acquired on a LTQ mass analyser. However, the peaks of the isolated precursor ions (m/z 835 and 981)   don´t displays the characteristic isotopic distribution of a chloride atom.

To collect the tandem mass spectra of the [M+Cl]- adducts, the isolation window width must be higher than 2 m/z.

Please, if possible, these spectra must be reacquired using the correct instrumental conditions.

Reviewer 3 Report

The manuscript ID: foods-2071348 by Yuan et al., entitled “Metabolomic approach for identification of Panax ginseng metabolites using internal extractive electrospray ionization mass spectrometry” describes the use of iEESI-MS analytical technique for detection of active substances from ginseng, as well as unsupervised multivariate statistical models for distinguishing samples of different origins. The technical soundness of the manuscript needs to be substantially improved, considering the following major issues that need to be addressed before further consideration:

 -    - The objectives are not clear. First, the authors declared the use of the iEESI-MS technique to identify metabolites in ginseng, which led them to bypass tedious chromatographic separations. Then, they described using the OPLS-DA model to demonstrate metabolic changes because of plants' different origins. On the other side, authors test different solvents to optimize the best one for detection. I think that it must be stated which are the linking points between these and rewrite the manuscript to improve the narrative line, which is even no obvious at all throughout the introduction, after which a reader could leave confused about the paper's hypothesis conducted.

-       Authors used a mass spectrometer with the ion-trap analyzer. It is well known that this is a low-resolution mass spectrometer (unitary resolution), which is limited to identifying substances. However, in this work, authors report unambiguously identification of metabolites, including their stereoisomers (R or S). According to my experience, achieving that without retention time (HPLC) and a high-resolution mass spec is quite hard. I suggest using the recommendation for reporting metabolite data in plant metabolomics by Fernie et al., 2011 (The Plant Cell, Vol. 23: 2477–2482).   

-       What are the analytical advantages of using iEESI-MS instead of LC-MS? Time of analysis? However, the authors are not considering the strong ion suppression phenomenon here, like in the direct infusion method, that substantially affects not only detection but also the dynamic range of their technique. At least, a referential LC-MS method should have been used for validation, since quality parameters such as robustness, reproducibility, sensibility, LOD, LOQ, etc, could give more hints about the proposed method suitability.

-       What is the reason why only negative ionization was performed??? How do the authors explain that, in negative ionization, 0.5 mM ammonium chloride in methanol gave higher signals intensity that, for example, solvents using 0.1% formic acid? Why did you not use methanol solvent with 0.1% formic acid?? Is it the solvent itself and not the modifier which made the difference??

-       Why did you use OPLS-DA model without exploring your data first with unsupervised models like PCA for example?

-       I think that the use of OPLS-DA model is, at least, inappropriate. The authors said that the separation observed in the scores plot clearly indicates chemical differences between groups. However, it is known that the OPLS-DA model aggressively forces the separation between groups. That’s why it is mandatory to perform its validation. The authors did not mention any validation method, and they should, although they gave R2Y and Q2 metric values. They report that values are suitable for the validation of model predictability. However, the Q2 value is meager (0.691), indicating poor consistency between predicted and original data. Thus, the model predictability seems to fail the validation trial.

-       Finally, the authors called their work a “metabolomics approach” while it is just a fingerprinting of ginseng chemical components. For metabolomics, the author should have handled and conserved their samples in a different way that ensures the quenching of metabolic activity. They also should have biochemically contextualized their results and, most important, used an analytical technique with a much higher detection coverage, like LC-MS, that allows a maximum detection of the metabolome. That’s why, currently, this technique is extensively used in metabolomics. I suggest using concepts better adapted to this work, like “metabolite fingerprinting” or similar.   

Further minor comments:

-       Use cursive for the Latin name of the plant, even if it is written in the title.

-       Introduction: LC or LC-MS are not methods, but analytical techniques.

-       Please explain what you meant with the following sentence in the Introduction: According to the principle of mass spectrometry, identification of the characteristic components of different ginsenosides is the premise for the identification of ginsenoside metabolites.

-       It should be briefly explained somewhere what is “ginseng under forest”.

-       Figures 2 and 3: please use absolute intensity in vertical axis for comparative purposes.

-       Page 6, first paragraph: What authors meant with “peaks were more chaotic”?

Reviewer 4 Report

This manuscript reports the main ginsenoside content of three ginseng extracts, two of them from different geographical origin. An own-developed ionization source for internal extractive electrospray ionization mass spectrometry is used.

The manuscript should be improved addressing the following issues:

1) In addition to sample preparation, please indicate which the advantages of this technique over other ionization methods are

2) A comparison, regarding quality and quantity, of compounds detected using this technique with those detected using other ionization techniques would be relevant.

3) Generalization of fragmentation patterns under each measurement conditions is enough, while explanation for different compounds as examples is unnecessary

4) Line 106: solvents currently used for ginsenoside extraction? Please rewrite this sentence

5) Lines 178 - 181: what does it mean "chaotic"? This sentence does not make sense (" ... from the second solvent than from the individual solvents")

6) Lines 181 - 192: where the samples extracted sequentially with every solvent? Please rewrite this paragraph

7) Table 1 footnote: "0.5 mM ammonium chloride in methanol" is repeated twice. Please explain the difference between "primary signal" and "quadratic signal" and "so on"

8) Line 243: please rewrite: "The iEESI−MS apparatus was performed on ginseng". It does not make sense.

Round 2

Reviewer 1 Report

Authors have now improved the introduction by incorporating objectives. Methodology has been now improved by clarification given about experimental design. Discussion is also improved. I will recommend improvement in English language.

Reviewer 3 Report

After reading answers to my comments, I think the manuscript has not been substantially improved. Remain fundamental technical issues that should be addressed.
For instance:
- You did not apport the complete, or at least primary, chemical data for identification of active substances. Just fragment ions obtained from a low-resolution mass spectrometer is too weak evidence for current standards.
- Regarding validation of this technique for detecting the substances of interest, rapid analysis can not be enough. Other soundness variables need to be reported, like the limit of detection, dynamic range, or the extraction yield from the tissues using relative quantification, at least.
- Regarding solvents and modifiers, no answer or further data was included. No supporting data for positive acquisition mode was reported, even if it was bad. In addition, why you did not use methanol with 0.1 formic acid is an unanswered question.
- Regarding the multivariate statistical model, you used OPLS-DA because separation in PCA was bad. Validation metric Q2 of the OPLS-DA model is too low because groups are not significantly different. Such overlaps in the PCA scores plot should gave you hints about this. Thus, your method may not be reproducible. Furthermore, you said that 0.4 is adequate for Q2, but no reference was cited. Indeed, it is quite low, indicating, once again, that groups are not significantly different. Therefore, the results are opposed to what you claimed.
